# Effect of dialysate buffer practices on serum parathyroid hormone concentrations in real-life french patients receiving hemodialysis

Pablo Antonio Ureña Torres[1,2]☯*, Moustafa Naja[3], Minh Hoang Tran[1,2],
Martine Cohen-Solal[4], Jordi Bover[5], Anne Kolko[1,2], Pascal Seris[1,2], Ibrahim Farah[1,2],
Natalia Alençar-de-Pinho[6], Ziad A. Massy[2,6], Charles Chazot[7]☯

1 Department of Nephrology and Dialysis, AURA Nord Saint-Ouen, Saint-Ouen, France, 2 Department of Renal Physiology, Necker Hospital, University of Paris Descartes, Paris, France, 3 AURA (Association pour l'Utilisation du rein Artificiel en Région Parisienne), Paris, France, 4 INSERM U1132 Université Paris-Cité and Department of Rheumatology, Lariboisière Hospital, Paris, France, 5 Department of Nephrology, University Hospital Germans Trias i Pujol (HGiTP), 8REMAR-IGTP Group, Research Institute Germans Trias i Pujol, Can Ruti Campus, Badalona (Barcelona), Catalonia, Spain, 6 Inserm Unit 1018, Team 5, CESP, Hôpital Paul Brousse, Paris-Saclay University (UPS) and Versailles Saint-Quentin-en-Yvelines University (UVSQ), Villejuif, France, 7 Clinique Charcot, Sainte Foy Les Lyon, France

☯ Contributed equally
* pablo.urena@aurparis.org; urena.pablo@wanadoo.fr

## Abstract

### Background

Dialysate composition is crucial for managing secondary hyperparathyroidism (SHPT) in patients on hemodialysis. Acetate-based dialysates elevate serum acetate and trigger adverse effects, while citrate and hydrochloric acid formulations might offer distinct clinical benefits.

### Methods

In this longitudinal observational real-life study, 876 adult patients on hemodialysis were classified by acid component of the dialysate (acetic acid, citric acid, or hydro-chloric acid). Monthly pre-dialysis parathyroid hormone (PTH) and bone-mineral biomarkers were measured. A linear mixed model (LMM), adjusted for baseline PTH, demographics, and clinical covariates, evaluated PTH trajectories over two years. The significance of the main effect of dialysate and its interaction with time was assessed using likelihood-ratio tests (LRT).

### Results

At baseline, serum PTH was significantly higher in the acetate group. The LMM confirmed significant influence of dialysate type (p < 0.001) and its interaction with time (p = 0.048) on PTH trajectories. Over two years, acetate showed a modest, non-significant PTH reduction of 2.9% (95% CI: −8.8% to +3.4%), whereas hydrochloric

**Data availability statement:** The study protocol, statistical analysis plan, and all data presented in this article are available to other researchers in the supporting material.

**Funding:** The author(s) received no specific funding for this work.

**Competing interests:** All the other authors declared no competing interests in relation with this study.

acid resulted in a significant cumulative PTH reduction of 10.9% (95% CI: −15.7% to −0.2%), with a significantly different yearly slope versus acetate (p = 0.044). Citrate showed a modest, non-significant change compared to acetate (−2.3%; 95% CI: −6.7% to +8.4%). Baseline PTH, phosphate, prior hypercalcemia, and dialysate calcium concentration were major predictors in the model.

## Conclusions

In adjusted longitudinal analyses, hydrochloric acid–based bicarbonate dialysate was associated with a steeper decline in pre-dialysis PTH over two years compared with acetate, while citrate did not differ from acetate. Prospective studies are needed to confirm whether buffer composition influences PTH control under standardized calcium-bath strategies.

## Introduction

Metabolic acidosis is common and requires effective management in patients on dialysis [1]. Acetic acid is the most used acid buffer in standard bicarbonate-based dialysates worldwide to prevent calcium and bicarbonate precipitation. However, the use of acetic acid-based bicarbonate dialysate (acetate) can increase serum acetate concentration by up to 5–10 times its physiological concentrations after a hemodialysis session [1–4]. Such acetate accumulation has been associated with the incidence of headaches, nausea, and per-dialysis hypotension [3–7]. Alternatives, such as citric or hydrochloric acid, have been proposed for acetate-free dialysis.

Citric acid-based bicarbonate dialysate (citrate) has been in use in France since 2013. It is safe at low concentrations and provides clinical advantages, including anticoagulant effects and a better biocompatibility compared to acetic acid [8]. Citrate dialysate reduces complement pathway activation, oxidative stress, and free radical formation, while improving mitochondrial glutathione synthesis [9–11]. Benefits also include fewer episodes of intradialytic hypotension, improved dialysis efficiency and nutritional status, better metabolic acidosis control, and decreased propensity for calcification [6,7,12–14]. However, citrate dialysate can lower serum magnesium levels, cause muscle cramps, induce a negative calcium balance, stimulate PTH secretion, and was initially suspected to increase mortality risk, though this has not been confirmed [2,8,15–17].

Hydrochloric acid-based bicarbonate dialysate, which for the sake of brevity will be referred to as hydrochloric dialysate, has been proposed as an alternative to acetate or citrate [3]. A study from the REIN registry conducted in 2015 indicated that acetate-free dialysis with hydrochloric dialysate was associated with improved survival in patients aged 70 years and older, regardless of comorbidities and the use of hemodiafiltration (HDF) [18]. However, the high acidity of concentrate acidified by hydrochloric acid introduces technical challenges in manufacturing, packaging, and use, which increase production costs.

Parathyroid hormone (PTH) is a central regulator of calcium and phosphate homeostasis. In CKD, elevated PTH leads to secondary hyperparathyroidism (SHPT), whereas low PTH results in adynamic bone disease. The type of dialysate used in hemodialysis (HD) can influence serum calcium and phosphate levels, thereby impacting PTH secretion [19]. However, there is no specific guidelines recommending a particular type of dialysate for patients receiving dialysis, likewise, there are uncertainties surrounding the effects of citrate dialysate on mineral and bone metabolism.

Beyond bicarbonate delivery, dialysate buffer composition may influence intradialytic ionized calcium through calcium binding and related shifts in calcium balance, which can modulate CaSR activation on parathyroid cells and thereby affect PTH secretion. Citrate can chelate calcium and may transiently lower ionized calcium, potentially stimulating PTH unless counterbalanced by a higher calcium bath, whereas acetate and hydrochloric acid–based dialysates may have different profiles. Accordingly, we hypothesized that dialysate buffer type could be associated with different longitudinal pre-dialysis PTH trajectories. Because intradialytic ionized calcium was not routinely available in this real-world dataset, we evaluated repeated pre-dialysis PTH measurements over follow-up.

The aim of this study was to examine the association between three types of bicarbonate-based dialysates (acetate, citrate and hydrochloric acid) and longitudinal changes in serum PTH and other mineral and bone biomarkers in routine practice.

## Methods

### Study design and population

This longitudinal observational real-life study primarily aimed to compare serum PTH levels among patients who were exposed to hydrochloric, acetate, and citrate dialysates over time. Secondary objectives included assessing the effect of these dialysates on pre-dialysis serum concentrations of bone-mineral metabolism biomarkers, including total calcium, phosphate, sodium, bicarbonate, total alkaline phosphatases (ALP). Additionally, the study evaluated the incidence of muscle cramps, arterial hypotension, hypomagnesemia, and both hypo- and hypercalcemia with each dialysate.

The demographic, clinical, biological characteristics as well as the outcomes were described for these three groups of adult patients (>18 years or older) on dialysis. Since acetate was the most widely used dialysate, it served as a comparator to describe the changes in serum PTH levels associated with the use of citrate and hydrochloric dialysates.

Four sites within the nephrology network, Association pour l'Utilisation du Rein Artificiel (AURA) in Paris, France, participated in the study: AURA Plaisance, which treated exclusively in-center patients and where hydrochloric acid dialysate was almost exclusively prescribed for historical reasons [20]; AURA Saint Ouen; ex AURA Corentin-Celton, and AURA Meaux. We included all patients who had PTH data between January 1st, 2022, and December 31, 2023. Non-opposition to patient-level data collection was obtained. The study was approved by the Institutional Review Board of Foch Hospital, Suresnes, France, registered under the number IRB00012437, and conducted in accordance with the ethical principles of the Declaration of Helsinki (S4 File).

Data were accessed for research on 18 July 2024. The analysis team worked on a pseudonymized dataset prepared by the data controller (AURA Paris); no direct identifiers were accessible during or after data collection.

### Data collection

Data collection was performed through patients' electronic health records (EHRs) using Medial software, Nantes, France. Pre-dialysis circulating bone-mineral metabolism biomarkers, including PTH, were extracted at routine monthly intervals. To ensure data accuracy and reliability, rigorous quality control procedures were implemented, such as verification of laboratory results and validation checks for data entry errors.

Medications, prescribed calcium and magnesium dialysate concentrations and adverse events were collected on a three-monthly basis. These were identified based on their documentation in the EHRs and categorized into binary variables (presence or absence) for analytical purposes. An event was considered present if it occurred at least once during

the specified time frame. Demographic data included the patient's site, Charlson comorbidity index, type of dialysate and medications. All the biochemical parameters collected were analyzed in the same laboratory at Saint Joseph Hospital in Paris. To examine changes in PTH over time, serum PTH levels were classified into three categories: Low: 0–120 pg/mL, within the recommended Kidney Disease: Improving Global Outcomes guidelines (KDIGO) target [21]: 121–600 pg/mL, and High: > 600 pg/mL.

Hypercalcemia was defined as a serum calcium level >2.65 mmol/L, and hypocalcemia, as a serum calcium level <1.80 mmol/L. Hypomagnesemia was defined as a serum magnesium level <0.80 mmol/L. Hypotension was defined as a mean systolic arterial blood pressure < 90 mmHg. These conditions were recorded if they occurred at least once during a HD session. Serum PTH concentration was assessed by an in vitro chemiluminescent microparticle immunoassay, the Architect Intact PTH assay from Abbott GmbH & Co. KG, Germany. Detailed protocol of the study can be found in S2 File).

## Statistical analysis

Categorical variables are presented as frequencies and percentages, and continuous variables as medians with interquartile ranges (IQR, Q1–Q3) due to their skewed distributions. Differences between two groups were evaluated using the Mann-Whitney U test, while differences among three or more groups were assessed with the Kruskal-Wallis test followed by Bonferroni-corrected pairwise comparisons. Categorical variables were compared using the Chi-square test.

We included all variables with <50% missing data and, assuming missing at random, applied multiple imputation by chained equations (MICE) to generate 20 imputed datasets, ensuring adequate relative efficiency of pooled estimates.

To assess the association between dialysate types (hydrochloric, citrate, and acetate [reference]) and PTH over time, we used linear mixed-effects models (LMMs). PTH was log-transformed to ensure residual normality. The primary inference on PTH evolution relies on the Dialysate×Time interaction (trajectory differences), whereas dialysate main effects reflect adjusted between-group differences at baseline/intercept. The models included random intercepts and slopes for time at the patient level to account for repeated measurements. An interaction term between dialysate type and time was tested using a likelihood ratio test (LRT) to evaluate different longitudinal PTH trajectories.

Models were adjusted for clinically relevant fixed-effect candidate covariates, including baseline PTH, age, sex, dialysis modality, dialysate calcium, biomarkers, and comorbidities but only significant were retained using backward elimination ($\alpha = 0.05$). Except for baseline PTH and demographics, covariates were incorporated as time-updated measures aligned with each PTH assessment (monthly biomarkers; and three-monthly medications, dialysate calcium prescription, and adverse-event indicators), rather than baseline values only. All model results from the imputed datasets were pooled according to Rubin's rules.

Adjusted differences in mean log-transformed PTH were back-transformed and are reported as percentage changes with 95% confidence intervals (CI). All statistical tests were two-sided, with significance set at $\alpha = 0.05$. Analyses were performed using Python (version 3.12.4) and RStudio (version 2024.09.1). Statistical analysis plan can be found in S3 File.

## Results

All analyses were conducted on 20 multiply imputed datasets. The total percentage of missing data for each variable is presented in Table S1 in S1 File. Imputed values closely matched the observed marginal distributions and the algorithm converged rapidly, demonstrating fidelity and stability (see S1 File). Key results are presented in Tables 1–4 and Figs 1,2.

## Patients

The study included 876 patients with a median dialysis vintage of 30 months (IQR: 5–81 months). At the baseline, 332 patients (37.9%) used hydrochloric, 292 (33.3%) citrate, and 252 (28.8%) acetate dialysates. The median age was 64 years (IQR: 51–73), and 559 (63.8%) were males (Table 1). Patients in the acetate group were younger (median age: 58 years, IQR: 45–69) compared to those in the hydrochloric (68 years, IQR: 55–76) and citrate groups (65 years, IQR:

**Table 1. Baseline main features of the whole study population.**

| | Acetate (n = 252) | Citrate (n = 292) | Hydrochloric (n = 332) | Total (N = 876) | p-value |
|---|---|---|---|---|---|
| Male, n (%) | 156 (61.9%) | 187 (64.0%) | 216 (65.1%) | 559 (63.8%) | 0.730 |
| Age (years), median (Q1–Q3) | 58 (45–69) | 65 (51–74) | 68 (55–76) | 64 (51–73) | <0.001 |
| Body mass index, kg/m$^2$, median (Q1–Q3) | 25.1 (21.6–29.0) | 25.4 (22.2–29.0) | 24.3 (21.3–28.9) | 25.0 (21.7–29.0) | 0.355 |
| Charlson comorbidity index, median (Q1–Q3) | 9 (4–10) | 10 (8–11) | 10 (9–11) | 10 (8–11) | <0.001 |
| Medical history, n (%) | | | | | |
| Hypertension | 219 (86.9%) | 240 (82.2%) | 285 (85.8%) | 744 (84.9%) | 0.259 |
| Diabetes | 72 (28.6%) | 105 (36.0%) | 139 (41.9%) | 316 (36.1%) | 0.004 |
| Cerebrovascular disease | 23 (9.1%) | 43 (14.7%) | 48 (14.5%) | 114 (13.0%) | 0.093 |
| Cancer | 20 (7.9%) | 25 (8.6%) | 48 (14.5%) | 93 (10.6%) | 0.015 |
| Hepatitis B | 22 (8.7%) | 24 (8.2%) | 16 (4.8%) | 62 (7.1%) | 0.122 |
| Hepatitis C | 6 (2.4%) | 14 (4.8%) | 16 (4.8%) | 36 (4.1%) | 0.261 |
| Hepatic cirrhosis | 3 (1.2%) | 5 (1.7%) | 11 (3.3%) | 19 (2.2%) | 0.176 |
| Peripheral artery disease | 36 (14.3%) | 56 (19.2%) | 68 (20.5%) | 160 (18.3%) | 0.140 |
| Type of nephropathy, n (%) | | | | | 0.065 |
| Diabetic | 32 (12.7%) | 61 (20.9%) | 73 (22.0%) | 166 (18.9%) | |
| Glomerulopathy | 41 (16.3%) | 39 (13.4%) | 39 (11.7%) | 119 (13.6%) | |
| Nephroangiosclerosis | 61 (24.2%) | 61 (20.9%) | 55 (16.6%) | 177 (20.2%) | |
| Renal Polycystosis | 18 (7.1%) | 23 (7.9%) | 28 (8.4%) | 69 (7.9%) | |
| Uropathy | 17 (6.7%) | 28 (9.6%) | 27 (8.1%) | 72 (8.2%) | |
| Other | 83 (32.9%) | 80 (27.4%) | 110 (33.1%) | 273 (31.2%) | |
| Type of dialysis, n (%) | | | | | <0.001 |
| Conventional hemodialysis | 203 (80.6%) | 169 (57.9%) | 138 (41.6%) | 510 (58.2%) | |
| Hemodiafiltration | 48 (19.0%) | 123 (42.1%) | 194 (58.4%) | 365 (41.7%) | |
| Hemofiltration | 1 (0.4%) | 0 (0.0%) | 0 (0.0%) | 1 (0.1%) | |
| Vintage (months), median (Q1–Q3) | 27 (6–83) | 38 (7–87) | 26 (4–68) | 30 (5–81) | 0.163 |
| Kt/V, median (Q1–Q3) | 1.30 (1.10–1.50) | 1.30 (1.10–1.55) | 1.30 (1.10–1.60) | 1.30 (1.10–1.50) | 0.490 |
| Dialysis duration (hours/session), n (%) | | | | | 0.873 |
| < 3 | 25 (9.9%) | 26 (8.5%) | 33 (9.9%) | 84 (9.4%) | |
| 3–4 | 224 (88.9%) | 273 (89.5%) | 297 (88.9%) | 794 (89.1%) | |
| > 4 | 3 (1.2%) | 6 (2.0%) | 4 (1.2%) | 13 (1.5%) | |
| Dialysate calcium concentration (mmol/L), n (%) | | | | | <0.001 |
| 1.25 | 1 (0.4%) | 1 (0.3%) | 8 (2.4%) | 10 (1.1%) | |
| 1.5 | 187 (74.2%) | 0 (0.0%) | 258 (77.7%) | 445 (50.8%) | |
| 1.65 | 0 (0.0%) | 212 (72.6%) | 0 (0.0%) | 212 (24.2%) | |
| 1.75 | 64 (25.4%) | 79 (27.1%) | 66 (19.9%) | 209 (23.9%) | |
| Medications, n (%) | | | | | |
| Calcimimetic | 69 (27.4%) | 60 (20.5%) | 65 (19.6%) | 194 (22.1%) | 0.057 |
| Calcium Salts | 77 (30.6%) | 77 (26.4%) | 103 (31.0%) | 257 (29.3%) | 0.391 |
| Phosphate Binders | 100 (39.7%) | 106 (36.3%) | 107 (32.2%) | 313 (35.7%) | 0.171 |
| Alfacalcidol | 83 (32.9%) | 65 (22.3%) | 64 (19.3%) | 212 (24.2%) | <0.001 |
| Native Vitamin D | 140 (55.6%) | 167 (57.2%) | 197 (59.3%) | 504 (57.5%) | 0.650 |
| Circulating bone-mineral biomarkers, median (Q1–Q3) | | | | | |

*(Continued)*

 

**Table 1.** (Continued)

|  | Acetate (n = 252) | Citrate (n = 292) | Hydrochloric (n = 332) | Total (N = 876) | p-value |
|---|---|---|---|---|---|
| PTH (ng/L) | 619 (351–967) | 443 (262–708) | 357 (187–611) | 451 (235–758) | <0.001 |
| Calcium (mmol/L) | 2.21 (2.08–2.33) | 2.21 (2.10–2.33) | 2.19 (2.09–2.31) | 2.20 (2.09–2.32) | 0.480 |
| ALP (mg/L) | 99 (75–139) | 91 (71–132) | 98 (73–133) | 96 (73–134) | 0.214 |
| Phosphate (mmol/L) | 1.42 (1.07–1.79) | 1.30 (0.96–1.67) | 1.31 (0.98–1.62) | 1.34 (1.00–1.68) | 0.014 |
| 25-OH-D | 37.9 (32.6–47.0) | 37.9 (30.0–44.1) | 37.8 (30.0–43.2) | 37.9 (31.0–45.0) | 0.119 |
| Other biomarkers, median (Q1–Q3) |  |  |  |  |  |
| Albumin (g/L) | 37.9 (34.6–41.1) | 36.3 (32.3–39.2) | 35.2 (32.0–38.7) | 36.5 (32.8–39.8) | <0.001 |
| Chloride (mmol/L) | 101 (98–104) | 102 (99–104) | 103 (100–106) | 102 (99–105) | <0.001 |
| Glucose (g/L) | 0.97 (0.79–1.37) | 1.12 (0.86–1.67) | 1.12 (0.91–1.73) | 1.09 (0.85–1.58) | <0.001 |
| Hemoglobin (g/dL) | 11.1 (10.2–11.9) | 10.7 (9.7–11.7) | 10.6 (9.5–11.6) | 10.8 (9.7–11.8) | 0.001 |
| Bicarbonate (mmol/L) | 23 (21–25) | 23 (21–25) | 22 (20–24) | 23 (21–25) | <0.001 |
| Sodium (mmol/L) | 138 (136–140) | 138 (136–139) | 137 (134–139) | 137 (135–139) | 0.003 |

Note: Baseline values presented in the table reflect raw data prior to imputation.

Abbreviations: Q1, first quartile; Q3, third quartile; CKD, Chronic Kidney Disease; ALP, Total Alkaline phosphatases.

Normal ranges: PTH 15–65 ng/L; Calcium 2.10–2.40 mmol/L; ALP 30–120 IU/L; BAP 15–41 mg/L; Phosphate 1.10–1.80 mmol/L; Albumin 35–50 g/L; Chloride 98–107 mmol/L; Glucose 0.70–1.10 g/L; Hemoglobin 10–11.5 g/dL; Bicarbonates 22–26 mmol/L; Sodium 135–145 mmol/L.

51–74). Significant differences were also observed in the Charlson Comorbidity Index (CCI) between groups (p < 0.001), with patients in the acetate group having a lower median CCI (9, IQR: 4–10) compared to the hydrochloric (10, IQR: 9–11) and citrate groups (10, IQR: 8–11).

Nephroangiosclerosis was more prevalent in the acetate group (24.2%) compared to citrate (20.9%) and hydrochloric groups (16.6%). Conversely, diabetic nephropathy was more common in the hydrochloric (22.0%) and citrate (20.9%) groups compared to the acetate group (12.7%). Conventional HD was predominantly used by patients in the acetate group (80.6%), whereas online HDF was more frequent in the hydrochloric (58.4%) and citrate groups (42.1%). The prevalence of diabetes (p = 0.004) and cancer (p = 0.015) was higher in the hydrochloric group than in the other groups. The acetate group had higher prescription rates of calcimimetic (27.4%) and alfacalcidol (32.9%) than the citrate (20.5% and 22.3%, respectively) and hydrochloric groups (19.6% and 19.3%, respectively). The median Kt/V and dialysis vintage was comparable across dialysate types.

Dialysate calcium concentration differed significantly across groups (p < 0.001). The concentration of 1.50 mmol/L was the most used: 77.7% of patients in the hydrochloric and 74.2% in the acetate group, but not used in the citrate group. Conversely, the concentration of 1.65 mmol/L was exclusively used in the citrate group (72.6%). The concentration of 1.75 mmol/L was used by 209 patients (23.9%) across all groups, distributed as follows: 25.4% in the acetate, 27.1% in the citrate, and 19.9% in the hydrochloric groups. The lowest concentration, 1.25 mmol/L, was rarely used (10 patients, 1.1%).

Table 2. Pooled LMM results examining the association between dialysate buffer composition and serum PTH levels. Acetate serves as the reference group (i.e., comparisons show changes relative to Acetate). Percentage changes in PTH (with 95% confidence intervals) are shown. Overall Dialysate and Dialysate×Time interaction p-values come from likelihood ratio tests (LRT); other p-values derive from Wald tests.

| Data & Fit | | | |
|---|---|---|---|
| Number of observations: 8519 Number of patients: 876 | | | |
| Residual σ² (Dispersion): 0.989 | | p-value: 0.78 | |
| Marginal R²: 0.538 | | | |
| Conditional R²: 0.795 | | | |
| Random Effects | | | |
| Patient level | | | |
| Component | Variance | SD | |
| Var (Intercept) | 0.195 | 0.442 | |
| Cov (Intercept, Time) | 0.082 | 0.286 | |
| Var (Time slope) | 0.054 | 0.233 | |
| Residual variance | 0.190 | 0.436 | |
| Fixed Effects | | % Change in PTH (95% CI) | p-value |
| Dialysate | | | <0.001 (LRT) |
| Citrate vs. Acetate | | +12.6% (+5.6% to +20.2%) | <0.001 |
| Hydrochloric vs. Acetate | | −9.3% (−14.6% to −3.6%) | 0.002 |
| Dialysate x Time (per year) | | | 0.048 (LRT) |
| (Citrate x Time) vs. (Acetate x Time) | | +0.3% (−3.4% to +4.1%) | 0.889 |
| (Hydrochloric x Time) vs. (Acetate x Time) | | −4.2% (−8.2% to −0.1%) | 0.044 |
| Medications | | | |
| Calcium salts | | −8.3% (−11.9% to −4.6%) | <0.001 |
| Alfacalcidol | | −8.6% (−12.0% to −5.1%) | <0.001 |
| Comorbidities | | | |
| Hypercalcemia before PTH | | −21.3% (−27.6% to −14.2%) | <0.001 |
| Continuous Covariates (per doubling) | | | |
| Baseline PTH | | +55.4% (+52.7% to +58.2%) | <0.001 |
| Phosphate | | +14.6% (+13.4% to +15.7%) | <0.001 |
| Calcium | | −5.9% (−6.8% to −5.0%) | <0.001 |
| ALP | | +9.3% (+7.9% to +10.7%) | <0.001 |
| Albumin | | +5.2% (+3.8% to +6.5%) | <0.001 |
| Sodium | | +1.8% (+0.9% to +2.6%) | <0.001 |
| 25-OH-D | | −1.5% (−2.6% to −0.4%) | 0.006 |
| Dialysate Calcium Concentration | | | |
| 1.65 mmol/L vs ≤ 1.5 mmol/L | | −5.8% (−11.8% to +0.6%) | 0.074 |
| 1.75 mmol/L vs ≤ 1.5 mmol/L | | −12.2% (−15.4% to −8.7%) | < 0.001 |
| Vintage (months) | | −0.01% (−0.03% to +0.01%) | 0.426 |

Throughout the study, 687 patients (78.4%) used only one type of dialysate, 184 (21.0%) used two types, and 5 (0.6%) used all three types. Among patients who switched dialysates, citrate was the first-line most commonly used (41.8%), with an average duration of use of 0.6 years and a change rate of 0.26 switches per patient-year, followed by hydrochloric (34.4%, average duration 0.6 years, change rate 0.20 switches per patient-year) and acetate (23.8%, average duration 1.0 year, change rate 0.16 switches per patient-year). Baseline PTH differed across groups (Table 1), reflecting non-random prescription; therefore, longitudinal analyses were adjusted for baseline PTH and other covariates.

**Table 3. Circulating mean bone-mineral biomarkers evolution for the three types of dialysates over time.**

| Biomarker | Dialysate | 0–3 months (mean±SD) | 9–12 months (mean±SD) | p-value over time* | p-values between groups at 9–12 months* |
|---|---|---|---|---|---|
| Calcium (mmol/L) | Hydrochloric | 2.21±0.14 | 2.23±0.13 | 0.119 | vs. Acetate: 0.049 vs. Citrate: 0.268 |
| | Acetate | 2.20±0.16 | 2.19±0.15 | 0.450 | vs. Hydrochloric: 0.049 vs. Citrate:>0.999 |
| | Citrate | 2.21±0.15 | 2.21±0.14 | 0.722 | vs. Hydrochloric: 0.268 vs. Acetate:>0.999 |
| Albumin (g/L) | Hydrochloric | 37.0±3.2 | 36.3±3.4 | 0.010 | vs. Acetate: 0.013 vs. Citrate: 0.757 |
| | Acetate | 37.9±4.2 | 37.4±3.8 | <0.001 | vs. Hydrochloric: 0.013 vs. Citrate: 0.198 |
| | Citrate | 36.6±4.1 | 36.4±3.7 | 0.366 | vs. Hydrochloric: 0.757 vs. Acetate: 0.198 |
| Phosphate (mmol/L) | Hydrochloric | 1.37±0.39 | 1.36±0.39 | 0.824 | vs. Acetate:>0.999 vs. Citrate:>0.999 |
| | Acetate | 1.43±0.51 | 1.43±0.53 | 0.675 | vs. Hydrochloric:>0.999 vs. Citrate: 0.605 |
| | Citrate | 1.38±0.44 | 1.37±0.43 | 0.501 | vs. Hydrochloric:>0.999 vs. Acetate: 0.605 |
| Sodium (mmol/L) | Hydrochloric | 137±3 | 137±3 | 0.171 | vs. Acetate: 0.013 vs. Citrate: 0.233 |
| | Acetate | 137±2 | 137±2 | 0.847 | vs. Hydrochloric: 0.013 vs. Citrate: 0.847 |
| | Citrate | 137±2 | 137±3 | 0.901 | vs. Hydrochloric: 0.233 vs. Acetate: 0.847 |
| ALP(mg/L) | Hydrochloric | 111±52 | 121±89 | 0.227 | vs. Acetate: 0.915 vs. Citrate: 0.727 |
| | Acetate | 133±116 | 130±96 | 0.253 | vs. Hydrochloric: 0.915 vs. Citrate: 0.091 |
| | Citrate | 103±55 | 103±47 | 0.804 | vs. Hydrochloric: 0.727 vs. Acetate: 0.091 |
| 25OHD | Hydrochloric | 39.0±10.4 | 42.2±11.3 | <0.001 | vs. Acetate: 0.288 vs. Citrate: 0.375 |
| | Acetate | 40.1±12.4 | 39.6±13.5 | 0.254 | vs. Hydrochloric: 0.288 vs. Citrate:>0.999 |
| | Citrate | 40.0±11.3 | 40.9±12.6 | 0.784 | vs. Hydrochloric: 0.375 vs. Acetate:>0.999 |

*Wilcoxon signed-rank for within-group over-time; Mann–Whitney for between-group at 9–12 months, with Bonferroni correction applied for multiple pairwise comparisons. Abbreviations: ALP, alkaline phosphatase. A p-value<0.05 was considered statistically significant.

### Serum PTH levels by dialysate types

**PTH category transitions over time.** Over the 24-month follow-up, the distribution of PTH categories showed notable divergence among dialysate types. Patients were assigned to the dialysate group used most frequently during each 3-month interval. At 0–3 months (n=260), 4% of patients using acetate had low PTH (<120 pg/mL), 45% were within the recommended range (121–600 pg/mL), and 52% had high PTH levels (>600 pg/mL). By 21–24 months (n=102), the proportion with low PTH remained unchanged (4%), within the recommended range increased modestly to 54%, and with high PTH decreased to 42%. Overall, these data indicate only minor improvements in PTH control among acetate users.

**Table 4. Annualized rate of adverse events per patient-year by dialysate type.**

| | Hydrochloric | Citrate | Acetate | p-value* |
|---|---|---|---|---|
| Hypomagnesemia before PTH, (events/patient-year) | 0.30 | 0.54 | 0.48 | 0.003 |
| Cramps before PTH,(events/patient-year) | 0.23 | 0.24 | 0.18 | 0.491 |
| Hypocalcemia before PTH, (events/patient-year) | 0.03 | 0.04 | 0.10 | 0.037 |
| Hypercalcemia before PTH, (events/patient-year) | 0.05 | 0.06 | 0.04 | 0.821 |
| Hypotension before PTH, (events/patient-year) | 0.22 | 0.27 | 0.13 | 0.019 |

*Chi-square test. A p-value of less than 0.05 was considered statistically significant.

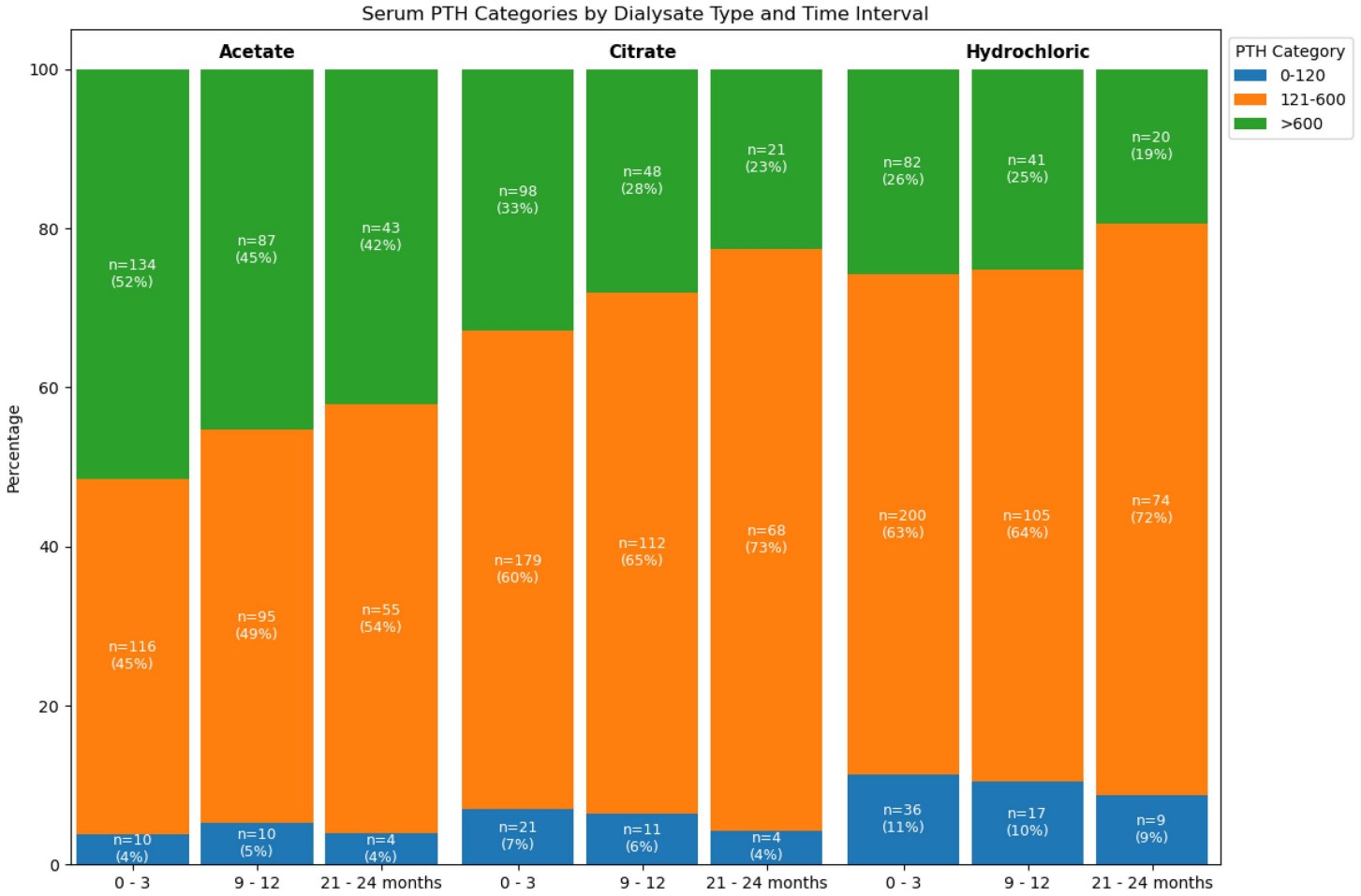

**Fig 1. Serum PTH levels by dialysate type over 24-month follow-up.**

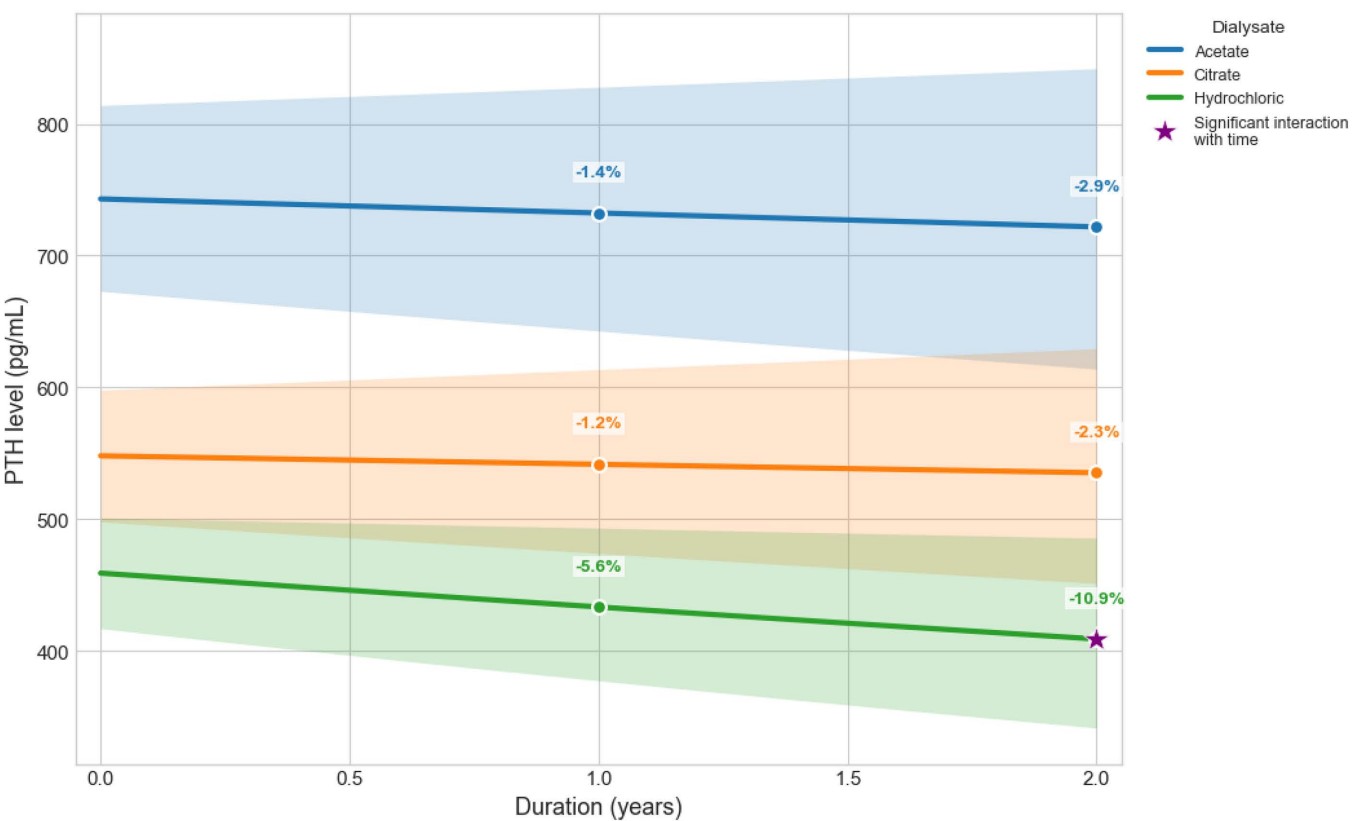

**Fig 2. Estimated PTH trajectories by dialysate for total patients.**

Among patients predominantly using citrate dialysate (n = 298 at 0–3 months), 7% had low PTH, 60% were within the target range, and 33% had high PTH. At 21–24 months (n = 93), the prevalence of low PTH decreased slightly to 4%, high PTH markedly declined to 23%, and patients within the target range increased substantially to 73%, demonstrating a significant improvement toward optimal PTH control.

Patients using hydrochloric dialysate (n = 318 at 0–3 months) initially had 11% low PTH, 63% within target range, and 26% high PTH. By 21–24 months (n = 103), the proportion with low PTH decreased slightly to 9%, high PTH reduced notably to 19%, and those within the recommended range rose to 72%, similarly reflecting meaningful progression toward optimal serum PTH levels (Fig 1).

**Analysis of serum PTH levels using LMM.** The linear mixed-effects model (LMM) to assess the associations between dialysate type (hydrochloric, citrate, and acetate as reference) and repeated serum PTH measurements, showed that the dialysate type significantly influenced serum PTH levels (LRT, p < 0.001), and the interaction between dialysate type and time was also significant (LRT, p = 0.048). Compared to acetate, hydrochloric dialysate was associated with a significant yearly reduction in PTH trajectory (−4.2% per year, 95% CI: −8.2% to −0.1%; p = 0.044). In contrast, citrate showed no significant difference compared to acetate (+0.3% per year, 95% CI: −3.4% to +4.1%; p = 0.889) (Table 2). Dialysis modality and predialysis bicarbonate were not significant and therefore were not retained in the final adjusted model.

Baseline PTH was strongly associated with subsequent PTH values, showing a 55.4% increase per doubling (95% CI: 52.7% to 58.2%; p<0.001). Similarly, significant associations per doubling were observed for phosphate (+14.6%, 95% CI: 13.4% to 15.7%; p<0.001), ALP (+9.3%, 95% CI: 7.9% to 10.7%; p<0.001), calcium (−5.9%, 95% CI: −6.8% to −5.0%; p<0.001), albumin (+5.2%, 95% CI: 3.8% to 6.5%; p<0.001), sodium (+1.8%, 95% CI: 0.9% to 2.6%; p<0.001), and 25-OH-D (−1.5%, 95% CI: −2.6% to −0.4%; p=0.006). Additionally, medication usage significantly influenced serum PTH levels: calcium salts were associated with a reduction (−8.3%, 95% CI: −11.9% to −4.6%; p<0.001), while alfacal-cidol usage also significantly decreased PTH (−8.6%, 95% CI: −12.0% to −5.1%; p<0.001). Higher dialysate calcium concentrations were also associated with lower PTH: compared to a concentration of ≤1.50 mmol/liter, use of 1.65 mmol/liter was linked to a non-significant 5.8% reduction in PTH (95% CI: −11.8% to +0.6%; p=0.074), whereas a concentration of 1.75 mmol/liter produced a significant 12.2% reduction (95% CI: −15.4% to −8.7%; p<0.001) (Table 2). Dialysis vintage in months was not significantly associated with PTH levels (p=0.426). The variance inflation factors for covariates in the LMM canbe seen in S2 Table in S1 File.

Fig 2 displays the estimated trajectories of PTH over two years, stratified by dialysate type as modeled by the LMM. While all dialysate types showed declining serum PTH levels, only hydrochloric dialysate demonstrated a statistically significant downward slope, achieving a cumulative decrease of −10.9% (95% CI: −15.7% to −0.2%) at two years. Citrate and acetate showed modest, non-significant declines of −2.3% (95% CI: −6.7% to +8.4%) and −2.9% (95% CI: −8.8% to +3.4%), respectively.

### Circulating bone-mineral metabolism biomarkers

Over the 9–12-month follow-up period, comparisons were made between mean baseline measurements (0–3 months) and subsequent mean values (9–12 months). Serum calcium levels remained stable within each dialysate group: hydro-chloric (2.21±0.14 to 2.23±0.13 mmol/L, p=0.119), acetate (2.20±0.16 to 2.19±0.15 mmol/L, p=0.450), and citrate (2.21±0.15 to 2.21±0.14 mmol/L, p=0.722). A modest but significant between-group difference was noted at 9–12 months between hydrochloric and acetate dialysates (p=0.049).

Serum phosphate and ALP showed no significant within- or between-group differences. Similarly, sodium levels were stable within and between dialysate groups, except for a modest between-group difference at 9–12 months between hydrochloric and acetate groups (p=0.013). 25-OH-D significantly increased over time in the hydrochloric group (39.0±10.4 to 42.2±11.3 ng/mL, p<0.001), while remaining stable in both acetate (40.1±12.4 to 39.6±13.5 ng/mL, p=0.254) and citrate groups (40.0±11.3 to 40.9±12.6 ng/mL, p=0.784). No significant between-group differences were observed at 9–12 months (Table 3).

### Adverse events

The annualized rate of hypomagnesemia events was significantly higher in the citrate group (0.54 events/patient-year) compared with hydrochloric (0.30 events/patient-year) and acetate (0.48 events/patient-year) groups (p=0.003). The incidence of cramps did not significantly differ between the three groups. Similarly, hypercalcemia rates showed no significant differences across dialysates: hydrochloric (0.05 events/patient-year), citrate (0.06 events/patient-year), and acetate (0.04 events/patient-year) (p=0.821). Conversely, significant differences were observed for hypocalcemia and hypotension rates. Hypocalcemia events occurred at 0.03, 0.04, and 0.10 events/patient-year for hydrochloric, citrate, and acetate dialysates, respectively (p=0.037). Hypotension events were reported at rates of 0.22, 0.27, and 0.13 events/patient-year in the hydrochloric, citrate, and acetate groups, respectively (p=0.019). These adverse-event rates constitute the main clinical correlates available in the electronic health record for this study (Table 4).

### Discussion

The main findings of this study are as follows: First, and for the first time in real-world setting, serum PTH levels significantly decreased over two years in patients exposed to bicarbonate-based hydrochloric acid dialysate, reaching a decline

of −10.9% at two years. In contrast, PTH declined only modestly and non-significantly by −2.9% and −2.3% in patients using acetate or citrate dialysates, respectively. Second, patients treated with citrate buffer did not display a rise in PTH but rather showed a small, non-significant downward trend over time, despite the interference of this buffer with divalent cations. Third, the three groups of patients differed markedly in baseline characteristics and treatment patterns, a case-mix that inevitably influenced the observed outcomes. Therefore, our findings should be interpreted as real-world associations between buffer type and PTH trajectories under routine co-prescription of different calcium baths and modalities, rather than as isolating the causal effect of buffer composition alone.

PTH is essentially regulated by circulating ionized calcium concentration, which binds and activates the calcium-sensing receptor, a G protein-coupled receptor present in parathyroid cells, to inhibit PTH production [21]. Acid–base status may also influence the calcium–PTH set-point and potentially modify responsiveness to calcimimetics. Although predialysis bicarbonate was available and described, intradialytic pH/ionized calcium dynamics were not captured, so residual confounding related to acid–base physiology cannot be excluded. Consequently, selecting an appropriate dialysate calcium concentration is crucial for maintaining neutral calcium balance and keeping PTH within the KDIGO-recommended values (2–9 times the upper normal threshold). The 2025 KDIGO controversy highlights persistent uncertainty over the optimal PTH target, as well as the absence of high-quality data informing the optimal calcium-dialysate concentration. Nevertheless, they suggest, based on observational studies, that using a dialysate calcium of 1.25 to 1.50 mmol/L would be the most appropriate concentrations in order to prevent CKD-MBD and vascular calcifications [22]. If the calcium balance becomes negative with these dialysates, this can be countered by prescribing a high-calcium diet, oral calcium salts, calcium-based phosphate binders, and/or active vitamin D analogues. However, these recommendations have never taken into consideration the type of acid component of the buffer used for the dialysate fabrication, which may have a direct impact on the dialysate free calcium concentration and its per-dialytic transfer to the patient. Our results support this assertion by demonstrating a significant dialysate-type × time interaction in the LMM (p = 0.048), supporting that dialysate buffer type is associated with different PTH trajectories over two years after multivariable adjustment.

One corroborating finding of our study is that PTH tended to decrease toward the recommended values in the three groups of patients. Although 25-OH-D increased modestly in the hydrochloric group, we cannot infer a causal or synergistic effect on PTH; the hydrochloric dialysate–associated decline in PTH persisted after adjustment for 25-OH-D and other covariates in the mixed-effects model.

Several mechanisms could underlie the greater decrease of PTH with hydrochloric dialysate. Differences in dialysate calcium prescription may partly contribute [23]; however, associations persisted after adjustment for dialysate calcium, suggesting buffer composition could play a role. The LMM nevertheless confirmed that dialysate-calcium concentration was an independent predictor of PTH: compared with ≤ 1.50 mmol/L, a bath of 1.75 mmol/L lowered PTH by 12.2% (95% CI −15.4% to −8.7%; p < 0.001), whereas 1.65 mmol/L produced a borderline 5.8% reduction (p = 0.074). Alternatively, hydrochloric dialysate could favor greater calcium transfer to patients, resulting in a more positive calcium balance; unfortunately, total calcium balance could not be assessed in this study. Of note, although the pre-dialysis serum total-calcium levels did not differ between the three groups, they accounted for 5.9% of the PTH variation in the hydrochloric group according to the LMM. The second possibility could be better PTH control because the baseline PTH was already within the recommended KDIGO range in 55% of these patients, which is a well-known determinant of PTH evolution [24]. The third possibility is that patients on hydrochloric dialysate were older, and had more comorbidities, including diabetes, conditions known to be associated with low PTH [25].

As citrate is a well-known calcium binder, its use as a dialysate buffer, even at relatively low concentrations (0.8–1.5 mmol/L), has been associated with a negative calcium balance and increased PTH [6,7,12–14]. In our study, we did not observe any significant increase of PTH; on the contrary, PTH decreased slightly by 2.3% after two years of follow-up. The proportion of patients with PTH in the high PTH category (>600 pg/mL) decreased from 33 to 23%, and there was a 13% increase in the proportion of patients with PTH within target at two years. This effect was probably related to the use of a

calcium dialysate concentration of 1.65 mmol/L in 72.6% of these patients. We found a higher incidence of hypomagne-saemia (0.54 events patient-year) in patients using citrate compared to hydrochloric acid (0.30) and acetate (0.48), with no difference in the incidence of muscle cramps, probably because of the high dialysate-calcium concentration. Citrate dialysate may offer other clinical advantages, including anticoagulation effects and better biocompatibility compared to acetic acid [8]. It reduces complement pathway activation, oxidative stress, and free-radical formation, while improving mitochondrial glutathione synthesis [9–11]. It is also associated with fewer episodes of intradialytic hypotension, improved dialysis efficiency, better metabolic-acidosis control, decreased propensity for calcification, and improved nutritional status [6,7,12–14]. Altogether, these data suggest that citrate dialysate, which has been in use in France since 2013, has not been associated with increased mortality in prior observational reports [15–17].

KDIGO guidelines recommend a calcium-dialysate of 1.25 or 1.50 mmol/L to avoid positive calcium balance and vascular calcifications [22]. Our data show that 1.50 mmol/L was prescribed in 50.2% of patients, whereas 1.25 mmol/L was used in only 1.1%. A strict restriction to a single calcium bath across all three dialysate types was not feasible due to local prescribing patterns (notably, 1.50 mmol/L was not used in the citrate group, and 1.65 mmol/L was exclusive to citrate), so calcium-bath imbalances may still contribute despite multivariable adjustment. Therefore, we cannot robustly address the effect of a 1.25 mmol/L bath on PTH dynamics. This point is clinically important, because large observational programs (e.g., DOPPS) have reported temporal increases in PTH following implementation of lower dialysate calcium recommendations [26]. The decline in PTH was observed in our study in the group using hydrochloric dialysate at 1.50 mmol/L calcium (77.0% of patients).

The observation of a significantly higher PTH at baseline in patients on the acetate dialysate compared to hydrochloric and citrate dialysates was not surprising. This was likely related to the nephrologists' background and prescription habits, in the absence of specific internal, national, and international guidelines. Acetate prescription was predominant in outpatients and self-care HD units treating younger patients with fewer comorbidities and less fear of the potential deleterious effects of acetate. Even at the relatively low concentration of acetate in the dialysate (3.0 mmol/L), acetate may still perturb intermediary metabolism and generate several metabolites, including acetoacetate and β-hydroxybutyrate, which might potentially induce several adverse effects such as headaches, nausea, per-dialysis hypotension, and ultimately stimulating interleukin-1 production and inflammation [3,5–7]. Notably, recent experimental studies have suggested that microbiota-dependent butyrate plays an important role regulating the skeletal action of PTH [27].

Several limitations should be acknowledged. First, several covariates had substantial missingness (up to 50%). Although we used multiple imputation by chained equations under a missing-at-random assumption and diagnostics (Figs S1–S2 in S1 File) supported convergence and distributional similarity, bias cannot be excluded if data were missing not at random. Second, this was an observational study and dialysate type was not randomly assigned; residual confounding, including confounding by indication, may persist despite adjustment. Third, a 1.25 mmol/L dialysate calcium bath was rarely prescribed (<1%), precluding assessment of its impact on PTH trajectories. Fourth, the prevalent nature of the cohort may introduce survivor bias and limit generalizability to incident patients. Fifth, we did not assess hard clinical outcomes, and medication exposure was captured as quarterly indicators rather than dosages, preventing evaluation of treatment de-escalation. Finally, intradialytic or post-dialysis ionized calcium measurements and total calcium balance were unavailable; FGF23 was not measured; and post-dialysis bicarbonate was not routinely recorded.

In this real-world cohort, hydrochloric acid–based bicarbonate dialysate was associated with lower PTH levels over 24 months compared with acetate after adjustment for baseline PTH, dialysate calcium, medications and other covariates. Citrate dialysate was not associated with a higher PTH trajectory or higher cramp rates under the calcium-bath strategies used in routine practice. Given the observational design, baseline case-mix differences and the lack of intradialytic ionized calcium/pH measurements, prospective controlled studies are needed to clarify the independent role of buffer composition and optimal buffer–calcium bath combinations for CKD-MBD management.

 

## Supporting information

**S1 File. Supporting information S1S2 containing: Table S1: Summary of missing data across variables.** Table S2: Variance inflation factors for covariates in the LMM. Fig S1: Imputed versus observed distributions of clinical variables. The red line represents the distribution of the observed data, and the black lines represent the distributions across the imputed datasets; the close overlap supports consistency between observed and imputed values. Fig S2: Mean convergence of imputed values across iterations. The x-axis represents the number of iterations; stability across iterations (no drift) supports convergence of the imputation procedure.
(DOCX)

**S2 File. Supporting information S3 containing: Detailed study protocol.**
(DOCX)

**S3 File. Supporting information S4 containing: Statistical analysis plan.**
(DOCX)

**S4 File. Supporting information S5 containing: Human Subjects Research Checklist.**
(DOCX)

**S5 File. Supporting information S6 containing: Anonymized database.**
(XLSX)

## Author contributions

**Conceptualization:** Pablo Ureña-Torres, Minh Hoang Tran, Martine Cohen-Solal, Jordi Bover, Pascal Seris, Natalia Alencar-de-Pinho, Ziad Massy, Charles Chazot.

**Data curation:** Pablo Ureña-Torres, Mustafa Naja.

**Formal analysis:** Pablo Ureña-Torres, Mustafa Naja, Natalia Alencar-de-Pinho, Charles Chazot.

**Funding acquisition:** Pablo Ureña-Torres, Charles Chazot.

**Investigation:** Pablo Ureña-Torres, Mustafa Naja, Minh Hoang Tran, Martine Cohen-Solal, Pascal Seris, Ibrahim Farah, Charles Chazot.

**Methodology:** Pablo Ureña-Torres, Mustafa Naja, Minh Hoang Tran, Natalia Alencar-de-Pinho, Ziad Massy, Charles Chazot.

**Project administration:** Pablo Ureña-Torres, Mustafa Naja, Ziad Massy, Charles Chazot.

**Resources:** Pablo Ureña-Torres.

**Software:** Pablo Ureña-Torres, Mustafa Naja.

**Supervision:** Pablo Ureña-Torres, Natalia Alencar-de-Pinho, Ziad Massy, Charles Chazot.

**Validation:** Pablo Ureña-Torres, Mustafa Naja, Minh Hoang Tran, Martine Cohen-Solal, Jordi Bover, Anne Kolko, Pascal Seris, Ibrahim Farah, Natalia Alencar-de-Pinho, Ziad Massy, Charles Chazot.

**Visualization:** Pablo Ureña-Torres, Mustafa Naja, Minh Hoang Tran, Martine Cohen-Solal, Jordi Bover, Anne Kolko, Pascal Seris, Ibrahim Farah, Natalia Alencar-de-Pinho, Ziad Massy, Charles Chazot.

**Writing – original draft:** Pablo Ureña-Torres, Mustafa Naja, Minh Hoang Tran, Martine Cohen-Solal, Jordi Bover, Anne Kolko, Pascal Seris, Ibrahim Farah, Natalia Alencar-de-Pinho, Ziad Massy, Charles Chazot.

**Writing – review & editing:** Pablo Ureña-Torres, Mustafa Naja, Minh Hoang Tran, Martine Cohen-Solal, Jordi Bover, Anne Kolko, Pascal Seris, Ibrahim Farah, Natalia Alencar-de-Pinho, Ziad Massy, Charles Chazot.

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
