## [Decision Letter · Decision Letter 0]

2 Dec 2025

Dear Dr. Ureña-Torres,

Thank you for submitting your manuscript to PLOS ONE. After careful consideration, we feel that it has merit but does not fully meet PLOS ONE’s publication criteria as it currently stands. Therefore, we invite you to submit a revised version of the manuscript that addresses the points raised during the review process.

The reviewers provided some mixed feedback about the manuscript’s methodology. However, the research question examined here remains insufficiently explored in the current literature, making the present study an important contribution and highlighting the need for dedicated prospective investigations and clinical trials. The statistical analysis used is appropriate but I kindly ask the authors to incorporate dialysis vintage into the adjusted model.

In addition, the limitations section should be expanded to acknowledge the substantial proportion of missing data for several variables (up to 50%), as well as the quasi-absence of patients dialyzed with a 1.25 mmol/L calcium bath. It should also be noted in the limitations that the cohort is composed of prevalent rather than incident patients, which may introduce survivor bias and limit the generalizability of the findings.

Please provide on top of these editorial recommendations, a point-by-point response to all reviewers.

We look forward to receiving your revised manuscript.

Kind regards,

Mabel Aoun, MD, MPH

Academic Editor

PLOS ONE

Journal Requirements:

2. In the online submission form, you indicated that “The study protocol, statistical analysis plan, and all data presented in this article are available to other researchers on request.”

4. Please amend the manuscript submission data (via Edit Submission) to include author Moustafa Naja, Minh Hoang Tran, Martine Cohen Solal, Jordi Bover, Anne Kolko, Pascal Seris, Ibrahim Farah, Natalia Alençar-de Pinho, Ziad A. Massy, Charles Chazot.

5. Please ensure that you refer to Figure 1 and 2 in your text as, if accepted, production will need this reference to link the reader to the figure.

Reviewers' comments:

Reviewer's Responses to Questions

**Comments to the Author**

1. Is the manuscript technically sound, and do the data support the conclusions?

Reviewer #1: No

Reviewer #2: Partly

Reviewer #3: Yes

Reviewer #4: Yes

Reviewer #5: Partly

2. Has the statistical analysis been performed appropriately and rigorously?

Reviewer #1: Yes

Reviewer #2: I Don't Know

Reviewer #3: I Don't Know

Reviewer #4: Yes

Reviewer #5: No

3. Have the authors made all data underlying the findings in their manuscript fully available?

Reviewer #1: No

Reviewer #2: Yes

Reviewer #3: Yes

Reviewer #4: Yes

Reviewer #5: Yes

4. Is the manuscript presented in an intelligible fashion and written in standard English?

Reviewer #1: No

Reviewer #2: Yes

Reviewer #3: Yes

Reviewer #4: Yes

Reviewer #5: No

Reviewer #1: The authors attempt to compare 3 different types of dialysate (hydrochloric, citrate, and acetate on mineral parameters. The main finding is a 10% decline in PTH in the hydrochloric acid group, whereas the citrate group showed a 10% increase after 24 months of follow-up. The major drawback of the study are 1) the substantial differences in the study population especially the HCl Dialysate group had the lowest PTH level at baseline, and 2) the lack of clinical outcomes associated with the decline in PTH level.

The less severe HPT can be more easily influenced by treatment or intervention compared to the more severe degree of PTH. The clinical outcomes on the decline in PTH levels such as a reduced dosage of calcimimetics or active vitamin D should be presented to confirm the significant benefit of HCl dialysate. In addition, the manuscript and all tables are difficult to follow.

Minor points

1.A calcium-dialysate concentration of 1.25 mmol/L was almost absent from clinical practice (<1 % of use). – How is this relevant to the study?

2.The progressive significant decrease of PTH in the hydrochloric dialysate was accompanied by a significant rise in 25-OH-D, which increased from 39.0 ± 10.4 ng/mL at the baseline to 42.2 ± 11.3 ng/mL at one year (p<0.001), whereas serum calcium and phosphate remained unchanged in all three groups. Taken together, these observations suggest a synergistic effect, whereby hydrochloric acid dialysate, combined with optimization of vitamin D status, drives superior biochemical control. – In dialysis patients, a change in 25-OH-D level is unlikey to affect PTH level.

Reviewer #2: The study addresses an important and clinically relevant question regarding the effect of dialysate buffer composition on PTH dynamics in HD patients. However, several major issues limit the interpretability of the findings and should be addressed before the manuscript can be considered for publication.

1. Introduction not sufficiently aligned with the study aim

The Introduction reviews the background on acetate, citrate, and hydrochloric dialysates but does not clearly articulate how dialysate composition affects calcium balance and PTH physiology, nor what differential effects the authors expected among the three groups. The rationale and the a priori hypothesis should be better specified, ideally summarizing the mechanistic links between dialysate buffer, ionized calcium, CaSR activation and PTH secretion. This clarification is essential to contextualize the results.

2. Significant differences in dialysis modality among groups

The dialysate groups differ substantially in dialysis modality distribution, with a much higher proportion of HDF in the citrate and hydrochloric groups, while conventional HD predominates in the acetate group. Since HDF is known to influence calcium handling and PTH regulation, this represents a major potential confounder. Although statistical adjustment is mentioned, such adjustments may not fully account for this imbalance.

A stratified or interaction analysis (Dialysate × HD vs. HDF) is strongly recommended, or at minimum, the impact of modality differences should be explicitly discussed as a limitation.

3. Differences in dialysate calcium concentration

Dialysate calcium concentration differs significantly among groups, with citrate patients predominantly receiving 1.65 mmol/L, while hydrochloric/acetate more frequently received 1.50 mmol/L. Since dialysate calcium concentration is one of the primary determinants of PTH, this difference may independently explain the observed results. Additional sensitivity analyses adjusting for dialysate Ca strata, or restricting analysis to patients with the same Ca bath, would improve interpretability.

Reviewer #3: Study has several strong points, notably the real-life observational design, large sample size, and two-year follow-up, which lend credibility to your findings. The use of centralized laboratory analyses and robust statistical methods adds to the reliability of the results. However, to further strengthen your study, consider addressing the biased case-mix distribution in more depth and discussing its potential impacts more thoroughly. Increasing the inclusion of incident dialysis patients and incorporating post-dialysis serum calcium measurements and total calcium balance would provide a more comprehensive understanding of mineral metabolism. Additionally, acknowledging potential unmeasured confounders will help clarify the validity of your conclusions. Overall, your research provides valuable insights into the management of SHPT and underscores the need for more controlled studies in this field.

Reviewer #4: I have carefully read the article entitled

“Effect of Dialysate Buffer Practices on Serum Parathyroid Hormone Concentrations in Real-Life French Patients Receiving Hemodialysis”

by Dr. Ureña et al., in which the authors perform a large-scale analysis of laboratory data to investigate whether different dialysate solutions have distinct impacts on the evolution of serum PTH levels in patients undergoing hemodialysis.

The authors conclude that the type of dialysate has a substantial influence on medium-term PTH evolution: the citrate bath worsened PTHi levels, whereas the hydrochloric acid bath led to a decrease, using the acetate-containing bath as the reference. These results are adjusted for other influential variables, such as the use of calcium-containing phosphate binders, prescription of cinacalcet or vitamin D, and also for dialysate calcium concentration, for which they also confirm the well-known suppressive effect of increasing dialysate calcium on serum PTH levels.

As the authors show, the literature on hemodialysis using hydrochloric acid–based dialysate is very limited, and in this sense I believe the study is of interest for publication in the journal. Overall, I do not think major changes are needed. I only suggest adding a few comments.

The authors collected a large longitudinal dataset of dialysis parameters and PTHi serum levels, and performed their analysis using a mixed-effects linear model (MLM) to account for within-patient changes over time and to ultimately analyze the effect of dialysate type on the PTH slope. Given that the authors present in a table the collected variables and the percentage of missing data, I understand that the MLM analysis incorporated the time-varying covariates corresponding to each PTH measurement. It would be helpful to add a sentence clarifying explicitly that covariates were not only used at baseline but were included throughout the entire follow-up. This would make the methodology and interpretation clearer.

I also miss a table summarizing the composition of the commercial dialysate solutions used, according to the dialysis technique employed. These could be presented in a supplementary table.

Some studies have reported that acidic pH influences the calcium–PTH set-point and could potentially alter the response to cinacalcet. Patients with more severe acidosis tend to present higher PTH levels and are less sensitive to the effects of cinacalcet and serum calcium. I believe this aspect should be mentioned in the discussion. The authors should state whether pre-dialysis serum bicarbonate was included as a covariate and whether it was significant. If it was not included, I believe it would be worthwhile adding it to rule out this potential confounder.

It also appears that the authors did not include the dialysis technique as an independent covariate, separate from the dialysate type and dialysate calcium concentration. If it was not significant, I think this should be stated in both the results and the discussion to clearly confirm that the dialysis technique itself did not have an independent effect.

In the limitations, the authors should address the overfitting and potential biases introduced by imputing missing values. A 50% missingness threshold per covariate is somewhat above what is typically recommended. They may rely on the Figure S1 plots to support that the imputed values appear to follow a distribution reasonably similar to that of the original data.

Figure S1. The authors should specify that the red line represents the distribution of the observed data, and the black lines represent the distributions of the imputed datasets. It might be helpful to add a brief comment noting that both distributions overlap closely and that the imputed values are consistent with the underlying model. Curves for 25OHD and for BMI presented some problems but this variable was not selected in final model.

Figure S2. For the convergence plots, the authors should indicate that the x-axis represents the number of iterations, and they should add a brief comment explaining that convergence was achieved for all variables, with stability across iterations and no drift, thus supporting the validity of the imputation procedure.

Reviewer #5: The authors have conducted a retrospective study on the effect of dialysate buffer practices on serum PTH levels in patients on maintenance hemodialysis (HD). May I mention the following:

1. The hypothesis of the study seems to be: Dialysate buffer affects serum PTH in HD patients. Serum PTH is mainly affected by calcium, phosphate, FGF 23 AND the treatment of hyperparathyroidism given to the patients. Please give the plausible explanation

2. Please mention pre- and post dialysis serum bicarbonate concentrations

3. To say that the type of dialysate buffer is affecting serum PTH level all above factors would need to be analysed with appropriate statistical analysis

.

Reviewer #1: No

Reviewer #2: No

Reviewer #3: **Yes:**Krishna BaradhiKrishna BaradhiKrishna BaradhiKrishna Baradhi

Reviewer #4: No

Reviewer #5: No

---

## [Author Response · Author response to Decision Letter 1]

10 Feb 2026

Responses to the Editor and Reviewers

To the Editor

The reviewers provided some mixed feedback about the manuscript’s methodology. However, the research question examined here remains insufficiently explored in the current literature, making the present study an important contribution and highlighting the need for dedicated prospective investigations and clinical trials. The statistical analysis used is appropriate but I kindly ask the authors to incorporate dialysis vintage into the adjusted model.

Thank you for this suggestion. We added dialysis vintage (months) to the fully adjusted linear mixed-effects model. Vintage was not associated with serum PTH (−0.01% per month, 95% CI −0.03% to +0.01%; p = 0.426) and did not materially change the dialysate effects or the dialysate×time interaction. The updated results are reported in the Results section (“Analysis of serum PTH levels using LMM”, page 8).

In addition, the limitations section should be expanded to acknowledge the substantial proportion of missing data for several variables (up to 50%), as well as the quasi-absence of patients dialyzed with a 1.25 mmol/L calcium bath. It should also be noted in the limitations that the cohort is composed of prevalent rather than incident patients, which may introduce survivor bias and limit the generalizability of the findings.

Agreed. We expanded the Limitations paragraph to cover (i) substantial missingness (up to 50%) and the use of multiple imputation under a missing-at-random assumption, (ii) the near-absence of 1.25 mmol/L dialysate calcium prescriptions, and (iii) the prevalent (non-incident) nature of the cohort and potential survivor bias. These points are now stated in the Discussion (Limitations, page 11).

Please provide on top of these editorial recommendations, a point-by-point response to all reviewers.

We look forward to receiving your revised manuscript.

Kind regards,

Mabel Aoun, MD, MPH

Academic Editor

PLOS ONE

Journal Requirements:

2. In the online submission form, you indicated that “The study protocol, statistical analysis plan, and all data presented in this article are available to other researchers on request.”

All PLOS journals now require all data underlying the findings described in their manuscript to be freely available to other researchers, either 1. In a public repository, 2. Within the manuscript itself, or 3. Uploaded as supplementary information.This policy applies to all data except where public deposition would breach compliance with the protocol approved by your research ethics board. If your data cannot be made publicly available for ethical or legal reasons (e.g., public availability would compromise patient privacy), please explain your reasons on resubmission and your exemption request will be escalated for approval.

Thank you for highlighting the PLOS Data Availability Policy. We have now prepared a fully anonymised dataset underlying the findings (with removal of direct identifiers and generalisation/suppression of key indirect identifiers to minimise re-identification risk under GDPR and French data-protection requirements). The anonymised dataset underlying the findings of this study is provided as Supporting Information files.

We agree and have revised our Data Availability Statement accordingly. The anonymised individual-level dataset underlying the results is already provided as Supporting Information and will be freely accessible upon publication, consistent with PLOS.

4. Please amend the manuscript submission data (via Edit Submission) to include author Moustafa Naja, Minh Hoang Tran, Martine Cohen Solal, Jordi Bover, Anne Kolko, Pascal Seris, Ibrahim Farah, Natalia Alençar-de Pinho, Ziad A. Massy, Charles Chazot.

We have amended the submission data (Edit Submission) to include the full author list: Moustafa Naja, Minh Hoang Tran, Martine Cohen Solal, Jordi Bover, Anne Kolko, Pascal Seris, Ibrahim Farah, Natalia Alençar-de Pinho, Ziad A. Massy, and Charles Chazot.

5. Please ensure that you refer to Figure 1 and 2 in your text as, if accepted, production will need this reference to link the reader to the figure.

Figures 1 and 2 are already explicitly cited in the main text at the relevant points (Results), using “(Figure 1)” and “Figure 2…”. We have double-checked to ensure each figure is referenced in-text for production linking.

No specific reviewer recommendations to cite particular previously published works were provided; therefore, no additional citations were added.

Reviewers' comments:

Reviewer's Responses to Questions

Comments to the Author

1. Is the manuscript technically sound, and do the data support the conclusions?

Reviewer #1: No

Reviewer #2: Partly

Reviewer #3: Yes

Reviewer #4: Yes

Reviewer #5: Partly

2. Has the statistical analysis been performed appropriately and rigorously?

Reviewer #1: Yes

Reviewer #2: I Don't Know

Reviewer #3: I Don't Know

Reviewer #4: Yes

Reviewer #5: No

3. Have the authors made all data underlying the findings in their manuscript fully available?

Reviewer #1: No

Reviewer #2: Yes

Reviewer #3: Yes

Reviewer #4: Yes

Reviewer #5: Yes

4. Is the manuscript presented in an intelligible fashion and written in standard English?

Reviewer #1: No

Reviewer #2: Yes

Reviewer #3: Yes

Reviewer #4: Yes

Reviewer #5: No

5. Review Comments to the Author

Reviewer #1: The authors attempt to compare 3 different types of dialysate (hydrochloric, citrate, and acetate on mineral parameters. The main finding is a 10% decline in PTH in the hydrochloric acid group, whereas the citrate group showed a 10% increase after 24 months of follow-up. The major drawback of the study are 1) the substantial differences in the study population especially the HCl Dialysate group had the lowest PTH level at baseline, and 2) the lack of clinical outcomes associated with the decline in PTH level.

The less severe HPT can be more easily influenced by treatment or intervention compared to the more severe degree of PTH. The clinical outcomes on the decline in PTH levels such as a reduced dosage of calcimimetics or active vitamin D should be presented to confirm the significant benefit of HCl dialysate. In addition, the manuscript and all tables are difficult to follow.

We agree on both points. Hard clinical outcomes were not available in this EHR-based dataset, and medication exposure was captured as quarterly presence/absence (no doses), so we could not assess treatment de-escalation; we now state this explicitly in the Limitations, and we report adverse-event indicators as the main available clinical correlates (Table 4).

We also acknowledge baseline imbalances, including lower baseline PTH in the HCl group (Table 1). To address this, our mixed-effects models were adjusted for baseline PTH and other clinically relevant covariates; however, residual confounding cannot be excluded. Finally, we improved readability by adding clearer in-text pointers to Tables 1–4 and Figures 1–2.

Minor points

1.A calcium-dialysate concentration of 1.25 mmol/L was almost absent from clinical practice (<1 % of use). – How is this relevant to the study?

Dialysate calcium concentration is a key determinant of calcium balance and PTH control, and current practice/guidelines often discuss 1.25 vs 1.50 mmol/L baths. In our cohort, 1.25 mmol/L was almost never prescribed (<1%), so we could not reliably evaluate whether a lower calcium bath modifies PTH trajectories or interacts with dialysate type. We therefore reported this as a limitation in the page 11 of the revised version of the manuscript.

2.The progressive significant decrease of PTH in the hydrochloric dialysate was accompanied by a significant rise in 25-OH-D, which increased from 39.0 ± 10.4 ng/mL at the baseline to 42.2 ± 11.3 ng/mL at one year (p<0.001), whereas serum calcium and phosphate remained unchanged in all three groups. Taken together, these observations suggest a synergistic effect, whereby hydrochloric acid dialysate, combined with optimization of vitamin D status, drives superior biochemical control. – In dialysis patients, a change in 25-OH-D level is unlikey to affect PTH level.

We agree. Although 25-OH-D increased slightly in the hydrochloric group (statistically significant), the magnitude was small and we cannot infer a causal or “synergistic” effect on PTH in hemodialysis patients. in our adjusted LMM (which includes 25-OH-D and medication indicators), hydrochloric dialysate remained associated with a significantly steeper decline in PTH over time, suggesting that the observed trajectory difference is not explained by changes in 25-OH-D alone. We removed the synergy interpretation and now report the 25-OH-D finding as an associated observation.

Reviewer #2: The study addresses an important and clinically relevant question regarding the effect of dialysate buffer composition on PTH dynamics in HD patients. However, several major issues limit the interpretability of the findings and should be addressed before the manuscript can be considered for publication.

1. Introduction not sufficiently aligned with the study aim

The Introduction reviews the background on acetate, citrate, and hydrochloric dialysates but does not clearly articulate how dialysate composition affects calcium balance and PTH physiology, nor what differential effects the authors expected among the three groups. The rationale and the a priori hypothesis should be better specified, ideally summarizing the mechanistic links between dialysate buffer, ionized calcium, CaSR activation and PTH secretion. This clarification is essential to contextualize the results.

Agreed. We revised the Introduction to better link buffer composition to calcium balance and PTH physiology (ionized calcium changes during dialysis, CaSR activation, and downstream PTH secretion). We also added our a priori expectation that citrate, via calcium chelation, could lower ionized calcium and stimulate PTH unless offset by a higher calcium bath, and we clarified that intradialytic ionized calcium was not measured in this real-world dataset (we therefore evaluate longitudinal pre-dialysis PTH trajectories).

2. Significant differences in dialysis modality among groups

The dialysate groups differ substantially in dialysis modality distribution, with a much higher proportion of HDF in the citrate and hydrochloric groups, while conventional HD predominates in the acetate group. Since HDF is known to influence calcium handling and PTH regulation, this represents a major potential confounder. Although statistical adjustment is mentioned, such adjustments may not fully account for this imbalance.

A stratified or interaction analysis (Dialysate × HD vs. HDF) is strongly recommended, or at minimum, the impact of modality differences should be explicitly discussed as a limitation.

We agree that dialysis modality differed across dialysate groups and could confound PTH dynamics. In line with your suggestion, we performed a sensitivity analysis testing whether dialysis modality (HD vs HDF) modified the dialysate effect over time (per year) by adding a Dialysate × time × modality interaction to the linear mixed-effects model. We found no evidence of effect modi

---

## [Decision Letter · Decision Letter 1]

10 Mar 2026

Effect of Dialysate Buffer Practices on Serum Parathyroid Hormone Concentrations in Real-Life French Patients Receiving Hemodialysis

PONE-D-25-51404R1

Dear Dr. Ureña-Torres,

We’re pleased to inform you that your manuscript has been judged scientifically suitable for publication and will be formally accepted for publication once it meets all outstanding technical requirements.

Kind regards,

Mabel Aoun, MD, MPH

Academic Editor

PLOS One

Additional Editor Comments (optional):

Reviewers' comments:

Reviewer's Responses to Questions

**Comments to the Author**

Reviewer #3: All comments have been addressed

Reviewer #4: All comments have been addressed

Reviewer #5: All comments have been addressed

2. Is the manuscript technically sound, and do the data support the conclusions?

Reviewer #3: Yes

Reviewer #4: Yes

Reviewer #5: Yes

3. Has the statistical analysis been performed appropriately and rigorously?

Reviewer #3: Yes

Reviewer #4: Yes

Reviewer #5: Yes

4. Have the authors made all data underlying the findings in their manuscript fully available?

Reviewer #3: Yes

Reviewer #4: Yes

Reviewer #5: (No Response)

5. Is the manuscript presented in an intelligible fashion and written in standard English?

Reviewer #3: Yes

Reviewer #4: Yes

Reviewer #5: Yes

Reviewer #3: Authors e expanded the

Discussion/Limitations in the page 11 to discuss this potential bias, acknowledge

unmeasured confounding, and clarify that our findings should be interpreted as real. Unfortunately authors do knowledge that post calcium levels are not available.

Reviewer #4: I think the work can be accepted for publication after the changes made in relation to the comments made by the reviewers.

Reviewer #5: The detailed replies to the reviewers comments are satisfactory and the modifications have been incorporated

.

Reviewer #3: **Yes:**Krishna BaradhiKrishna BaradhiKrishna BaradhiKrishna Baradhi

Reviewer #4: No

Reviewer #5: No

---

## [Editor Report · Acceptance letter]

PONE-D-25-51404R1

PLOS One

Dear Dr. Ureña-Torres,

I'm pleased to inform you that your manuscript has been deemed suitable for publication in PLOS One. Congratulations! Your manuscript is now being handed over to our production team.

Kind regards,

on behalf of

Pr Mabel Aoun

Academic Editor

PLOS One